# A Personal Scientific Journey—Looking Back at My Journey in Science: How DNA Damage and Repair Led to the Role of Gap Junctional Intercellular Communication, Mechanisms of Tumor Promotion, Human Adult Stem Cells, “Cancer Stem Cells”, Two Types of “Cancer Stem Cells” and the Modulation of Human Diseases by Epigenetic Toxins/Toxicants, Nutrition and Diets

**DOI:** 10.3390/cancers17162647

**Published:** 2025-08-13

**Authors:** James E. Trosko

**Affiliations:** Department of Pediatrics & Human Development, College of Human Medicine, Michigan State University, East Lansing, MI 48824, USA; trosko@msu.edu

**Keywords:** adult stem cells, epigenetic mechanisms, multi-stage, multi-mechanisms of carcinogenesis, tumor promotion, cancer stem cells, gap junctions, bioethics

## Abstract

A young scientist’s must realize that their scientific career’s journey is never a straight path from his/her initial dream to finishing that career. Their talents, skills and motivations; encountering external historic events; and barriers and unforeseen opportunities will affect that path. I provide some factors that shaped my scientific achievements. Starting at a young age, not knowing what a scientific career was, I was influenced by the consequences of the uses of the atomic bombs in Japan. Nurtured by a caring high school teacher, I was fortunate, lucky and persistent enough to survive challenges that forced me to change my approach to preventing and treating cancer and other human diseases. During that journey, I was able to make discoveries in different disciplines and, in the end, about not only the pursuit of truth but the ethical choices a scientist must make to use or not use that knowledge.

## 1. Introduction

###  An Individual Perspective and History of How DNA Repair and Mutagenesis, as Well as the Discovery of Cisplatin, Led to Epigenetics, Human Stem Cells, Gap Junctions and Two Types of Cancer Stem Cells

Rarely, if ever, is the scientific solution to a natural problem a linear pathway. The journey is filled with *detours*, *curves*, *potholes* and *do not enter*, *wrong way*, *slow down*, *slippery when icy*, *road under construction*, *road ends* and *no way to get there from here* signs. In addition, one might have a clear destination in mind for their journey, but the ultimate destination may be nowhere near what was originally planned.

In addition, with one discipline under one’s belt, the original guiding paradigms, specific concepts, experimental techniques that were available, experiment limitations and understanding of potential mechanisms of the pathogenesis of a biological problem not known make the journey a torturous one. These problems are overarched by historical, political and economic influences. As one example of how one’s scientific contributions might be accepted or not by the scientific community is illustrated with this quote: “*The important part of a scientific discovery in almost any aspect of science is the reception it receives*, *and this is in large measure a social phenomenon. Whether a theory is accepted*, *rejected*, *ignored*, *attacked or deferred for later consideration is a question of how people behave*—*and behavior is not a logical problem*” [1].

In my case, back in 1960, when I entered my young scientific career as a radiation geneticist, the public, political leaders, funding agencies and scientists were focused on the effects of atomic bomb radiation on human health. The prevailing concern was that radiation-induced DNA damage in cells of exposed living organisms would lead to mutations, and then to death or various diseases, such as cancer. The major driving force was the linear no-threshold (LNT) hypothesis [2]. However, while I was involved as a graduate student doing radiation research on X-ray-induced mutations in Drosophila, I got side-tracked with the original discovery by Dr. Barnett Rosenberg of the world’s most widely used anti-cancer drug, cisplatin [3]. I finished my Ph.D. in radiation genetics. and was awarded a postdoctoral fellowship at Oak Ridge National Laboratory to continue my training in radiation biology, radiation-induced DNA damage and mutagenesis.

While at Oak Ridge National Lab as a postdoctoral student, I worked with the late Drs. R.B. Setlow, Sheldon Wolff and E.H.Y. Chu on measuring DNA damage, chromosome breaks and in vitro mammalian mutagenesis for three years. After leaving Oak Ridge National Laboratory to assume my first academic job as an Assistant Professor at Michigan State University, I reignited my past relationship with Dr. Barnet Rosenberg and his quest to understand how this heavy metal anti-cancer drug worked. In trying to get this drug approved by the USA Food and Drug Administration, for the use of a heavy metal chemical on human beings, it had to be shown that it was not mutagenic. I demonstrated that cisplatin, although it seemed to kill cancer cells in mice, was NOT a genomic DNA-damaging compound, and did not induce mutations in the exposed surviving cells. That was my first road block. My mindset was shaped, when I was a postdoctoral fellow at Oak Ridge National Laboratory, by my original discovery that ultraviolet (UV) light radiation could induce specific DNA lesions in the DNA of human cells, which could be repaired by normal cells, but not in cells from sunlight-induced skin cancers in the exposed skin of individuals with the skin-cancer-prone human genetic syndrome xeroderma pigmentosum [4,5]. It was later shown that the cells surviving the ultraviolet exposure had a higher frequency of mutations than the normal cells [6,7]. Naturally, this led me to examine other human syndromes that were predisposed to cancer, believing that the “driving force” was genomic DNA damage, lack of DNA repair, and higher mutations in the survivors of the radiation. However, that chosen road was negative because most of these human cancer-prone syndromes had normal DNA repair. How could the cancers appear in these cancer-prone individuals if their genomic DNA repair was “normal”?

## 2. The First Road Block Pushes Me into the Path of Epigenetics, the “Tumor-Promoting Stage” of Multi-Stage Carcinogenesis and Human Adult Stem Cells

Stepping back from this road would not get me to where I wanted to go. I was now at the McArdle Laboratory of Cancer Research at the University of Wisconsin—Madison in the laboratory of the late Dr. Van Rensselaer Potter and the giants in the field of “Chemical Carcinogenesis” as an invited National Cancer Institute Career Development awardee. Here, I was surrounded by an interesting, but not yet fully appreciated, finding that a “*promoting*”-, but not “*initiating*”-, chemical accelerated the appearance of skin cancer in mice. Working with the lead scientist and his student, we tested a hypothesis, published about the time I arrived there, that this tumor-promoting natural chemical, phorbol ester or TPA, worked by inhibiting DNA damage, leading to mutations in the surviving cells exposed to UV radiation [8]. To make a long story short, since I was a “DNA repair man”, we tested this hypothesis only to show that phorbol esters did not inhibit DNA damage. Another road sign, “*turn around, you are on the wrong road*” showed up. It turns out that in testing this hypothesis, we did the in vitro assay incorrectly, and when we examined the nature of our experimental technique to measure mutations in the exposed cells, we noticed the design had us use too many cells, such that they were touching each other when phorbol ester was added.

Why was this to be so important? I tried to find out, though I never heard of gap junctions, or that cancer cells lacked gap junctional intercellular communication, unlike “normal cells”, which had gap junctional intercellular communication [9,10]. To this day, I am not sure how I found the early work of Loewenstein & Kanno, but it seems like there was an “*off ramp*” sign to test why “normal” and “cancer” cells, which were sensitive to this non-genomic DNA-damaging chemical, could “*promote”* but not “*initiate*” or mutate cells. When tested, it was clearly shown that phorbol ester did not damage genomic DNA, did not inhibit the repair of genomic damaged DNA, but did, reversibly, inhibit gap junctional intercellular communication [11]. Now, it seems, I was on the right path. Aided by many other studies published both by those in the gap junction field and some in the cancer and radiation fields, it seemed so obvious to me, but apparently not many others, that the road ahead, while not smooth, was going in the right direction [12].

Unlike many other investigators, my being in the late Dr. Van Potter’s lab gave me a new view of what was the one individual cell of the human body—with over 200 trillion cells—that was initiated by a process that converted it to a unique, but not “normal”, cell, while also not yet a malignant, invasive cancer cell with all the so-called “Hallmarks of Cancer” [13,14]. At that time there were two completely opposing hypotheses of the start of the multi-stage, multi-mechanism process of carcinogenesis, namely, the “Stem Cell Hypothesis” [15] and the “De-Differentiation” or “Reprogramming Hypothesis” [16]. While the concept of “stem cells” was an early idea for developmental, cell and a few cancer biologists, it was an “abstract” idea, not based on any real molecular characterizations of a single isolated human embryonic or adult stem cell. That was until papers appeared showing the isolation of human embryonic stem cells [17,18] and, later, the amazing demonstration that one could isolate, by “reprogramming” from a population of somatic differentiated human fibroblasts, an “induced pluri-potent stem cell” (“iPS” cell) [19]. While widely accepted by the scientific community, with a Nobel Prize to Dr. S. Yamanaka, and repeated, experimentally, by many individual labs and commercial companies, I believed there might be an error in the interpretation of the origin of those few “iPS” cells in the population of early skin explants.

That flaw in interpretation is that an early skin fibroblast biopsy leads to a population of cells including a few adult fibroblast stem cells that exist in that population. Another explanation is that transfecting both differentiated somatic fibroblasts and rare fibroblast stem cells with Yamanaka factors may result in isolating a few iPS cells. The reason is because these endogenous fibroblast adult stem cells have their normal endogenous embryonic genes, including OCT4A, and the exogenous copies of the Yamanaka genes. The differentiated fibroblasts with only the exogenous Yamanaka genes would not survive, since they were not “re-programmed” [20,21]. This is not surprising, as decades of trying to immortalize normal human fibroblasts have failed [22,23]. It must be noted that normal human fibroblasts have about 50 passages before senescence [24]. This is most likely because the fibroblast population in the early explant in vitro contains fibroblast stem cells [25,26]. The ultimate senescence of the culture after 50 passages is due to the loss of those few stem cells. This interpretation was supported by the experimental fact that my lab’s journey had isolated, for the first time, organ-specific adult stem cells that were shown to express the Oct4A gene but not have any GJIC function, while at the same time not expressing the connexin genes [27]. Further, using human adult breast stem cells, our lab showed that only these adult stem cells could lead, by sequential experimental treatment (exposure to SV40 virus Large T genes, radiation treatment, followed by transfection with the Neu/ERB2 gene), to a metastatic invasive breast tumor cell [28].

## 3. A New Road Sign Opened up an Unforeseen Characterization of Normal Human Stem Cells and Different Kinds of “Cancer Stem Cells”

While there were a few more “*slippery when icy*” signs on my journey, one that caught my attention was the observation that there were some “cancer stem cells” that did have expressed connexins but did not have functional gap junctions. This brought back some earlier research I had done on normal GJIC-positive cells, that when transfected with various oncogenes, such as H-ras, K-ras, neu, SV40, etc., would not have functional gap junctions and not express the OCT4A gene, yet would have the ability to become tumorigenic [29].

We can stop on our journey and look at the “*Big Picture*”: what did the scientific cancer community know at this time? First, all tumors appear to be a mixture of “cancer stem cells”, “cancer non-stem cells”, stromal fibroblasts and immune cells [30]. Also, cancer stem cells seem to be resistant to radiation and chemical toxins/toxicants by virtue of having better DNA repair and better anti-oxidant activity than the “partially-differentiated cancer non-stem cells” and other normal non-cancer cells in the mixture [31]. This complex mixture of different cell–cell interactions (soluble signals, such as growth factors, hormones and cytokines) and different intracellular pathway signals for growth control, proliferation and cell death, as well as the lack of gap junction function by two different means—one by no connexin gene being expressed and the other by having connexin genes expressed but rendered non-functional by some oncogene—makes it impossible to kill the “cancer stem cells” with one anti-cancer treatment [32,33].

Even multiple treatments have not eliminated cancer. Conceptually, if the original hypothesis by Borek et al. [34]. was correct, the universal phenotype of cancer (cancer stem cells) was its inability to have growth control, inability to terminally differentiate and inability to die by apoptosis. This inability to senesce by having the ability to remain “immortal” or not to “moralize” was associated with gap junction function. Remember, stem cells are naturally “immortal” until they are induced to differentiate or become “mortal”. From the observation that there seems to be two types of “cancer stem cells”—one with its connexin genes never having been expressed during the entire multi-stage, multi-mechanism process of carcinogenesis and the other type having expressed its connexin genes but also having expressed oncogenes, which prevent the connexin proteins from becoming functional [35]—then, conceptually, no one anti-cancer treatment will affect both types. For the first type, one should find agents that can induce the expression of the connexin gene(s) so the cancer stem cell will terminally differentiate or die by apoptosis. For the other “cancer stem cell type”, one must find ways to render the inhibition of the active oncogene product, to free the connexin protein to make functional gap junctions [36].

## 4. Nearing the End of My Scientific Journey, It Was Time to Assess Progress to Date and to Plot the Next Path

This now allows me to look in the rear-view mirror at my 50+-year journey from the discovery of cisplatin, to the discovery of how epigenetic chemicals modulate gap junction function, to the isolation of human organ-specific adult stem cells and their use in three-dimensional in vitro assays to screen for chemo-preventive/chemo-therapeutic agents against cancer [37]. What I see, looking back, is a road sign that seemed so obvious now, but not when I first saw it over 50 years ago. Today, regardless of any cancer treatment, at best, the treatment is only to treat cancers as chronic diseases, not to try to find a potential “cure”. Yet, today, there seems to be one anti-cancer drug, acting on one gender, and on one specific type of cancer, that might be considered one that can be classified as a “curable” treatment. Cisplatin, given to a patient with testicular cancer, such as was seen in the international bike racer Lance Armstrong, caused 100% of his cancer cells to disappear, without evidence of cytotoxic necrotic cell death; either terminal differentiation or apoptosis of that germinal-type testicular cancer seemed to have occurred. If he had had another non-testicular type of cancer, such as a lung cancer, the same cisplatin treatment would have, at best, only given him an extension of his life span, not a “cure”. This implies that, in this testicular cancer, cisplatin did not damage the genomic DNA of these cancer cells but induced, epigenetically, pathways that turned on genes that caused either terminal differentiation or apoptosis in those germinal cancer stem cells.

## 5. Conclusions

In summary, my 50+-year scientific journey was clearly not a linear path to my destination, suggesting that in the classic “nature and nurture” concept of cancer (or any disease), genes play the predominant factor (nature-wise) in probably less than 10% of all cancers, and the other 90% of factors are environmental (nurture-wise) influences. Indeed, alcohol intake, over-exposure to sunlight or inhaled smoke-related chemicals cause most preventable cancers, while nutrition, diet and lack of exercise are the other major influencers of cancer [38]. Most interestingly, even in human inherited mutated connexin genes, many pathologies, including some cancers, have been noted [39], and the environmentally influenced cancers have strong links to the modulation of gap junction function in many of cancers and other diseases, such as birth defects, cardiac dysfunctions, atherosclerosis, and reproductive, immune and neurological dysfunctions (Google search *gap junctions and human diseases*). How, in my mind, have the greater non-gap-junction fields not yet fully integrated these observations?

I am about to end my journey in the field of epigenetics, human stem cells and gap junctions. However, working with Dr. Mari Dezawa of Tohoku University in Japan, she and her team have made a new link through gap junctions to control the symmetric/asymmetric cell division of her unique human stem cells, which might help in the stem cell therapy area by enabling the propagation of needed stem cells for therapeutic use [40].

I would be remiss if I did not mention that the pressure on young researchers in the biomedical fields to utilize artificial intelligence (AI) and highly sophisticated technologies that can generate unbelievable amounts of raw, “unbiased” data. These young scientists often do not have backgrounds in the philosophy and history of science, nor in the many interacting disciplines involved in all human disease pathologies, including evolutionary, psychological, social, cultural, economic, political and historical factors. A broader concept is needed, such as one in which a complex interaction affecting the sustainability or adaptive evolution must occur, as in a Global One Health approach [41,42,43]. The power of evolution, in all forms—physical, chemical, biological and cultural—has, as in the beginning during the transition from unicellular to multicellular forms of life, been observed experimentally [44]. The role of evolution has been largely ignored in understanding the human condition in human medicine [45]. Recently, the powerful role of evolution in tying together stem cell biology, gap junctional intercellular communication and homeostatic control during normal or abnormal development has been seen [46].

One of those biological factors related to the transition from single-cell organisms to multicellular organisms came about during the evolution of the oxygenation of atmospheric and aqueous environments [47,48]. The formation of the stem cell, having the ability to switch between symmetrical and asymmetric cell division depending on the level of oxygen, forced the creation of a “stem cell niche” [49]. When the oxygen level is low, stem cells remain as stem cells. However, if the oxygen level increases, the stem cells will either differentiate or undergo apoptosis [50]. In other words, stem cells are very sensitive, as are some of their molecular biomarkers, such as the Oct4A gene, to high levels of oxygen, since Oct4 gene expression is redox-regulated [51]. Growing stem cells under low-oxygen conditions allows them to proliferate longer than the so-called “Hayflick limit” [52,53].

In addition, there has been more recent emphasis on the role that the senescence of individual cells plays in the aging process [54], such that these senescent cells—some refer to them as “zombie” cells [55]—are producing factors that can either be detrimental to tissues or possibly even beneficial [56]. Here, the concept of “epigenetics” provides an insight into the distinction between “senescent cells” and aged or terminally differentiated cells that has been made [57]. Choi and his co-workers noted that when senescent cells are placed on de-acellularized subcellular matrices of young tissue, they “regain” the phenotype of young cells. On the other hand, if young cells are placed on acellularized matrices of senescent cells, they are induced to become senescent cells.

With the rise of artificial intelligence, first developed in the development of computer science by Alan Turing, it is befitting to end my commentary of the role of cell–cell communication in normal and abnormal development by pointing out Turing’s only paper applying his algorithm to biological systems of pattern formation. He utilized only biological concepts at the time, namely, how *extracellular* signaling triggered *intracellular* pathways in cells to alter the activity of the cells [58]. He depended on what the father of modern physiology (Claude Bernard) provided as the basis of hormone and growth factors mechanism of action and the basis for “homeostasis”. Therefore, while Turning’s algorithm does provide a useful, but not completely accurate, tool, it does not show a more integrated picture of how cells actually communicate in a multicellular organism. Gap junctional inter-cellular communication had not yet provided a more integrated system in a three-dimensional multicellular organism. That is, various extracellular molecules (e.g., extracellular matrices, secreted molecules such as hormones, growth factors and cytokines/chemokines) can, upon binding to various receptors on cells, trigger various intracellular signaling pathways in the targeted cells to (a) cause the cells to modulate gene expression and produce molecules that can either send positive or negative secreted molecules to the original signaling cell or some other cell; (b) modulate gap junction function between contiguous cells. This is an integrated cell-communication system that needs to be accounted for in a modified Turing algorithm. This is only now being appreciated [59,60].

This story of “What is cancer and how can we prevent and treat this disease?” is not yet finished. A new “fork in the road” faces us. One of the intriguing new challenges is the emergence of early-onset cancers [61,62,63]. This emergence is world-wide and suggests complex global physical, chemical, ecological, behavioral, nutritional, dietary, cultural, economic, political and plural philosophical, religious and ethical factors.

### While the Original Goal on My Journey Has Been Reached, the Path Ahead Is Open for Others

All these interacting concepts, experimental findings and new technologies are seemingly linking evolution, stem cells, oxygenation altering epigenetic regulation of the homeostatic regulation of stemness or cell differentiation, and the importance of cell–cell communication via gap junctions or secreted factors. Hopefully, young scientists can now learn from all previous scientists that, as Yogi Berra, the famous baseball catcher, once noted in one of his many quotes, “*In life, everyone comes to a fork in the road*, *Take it**”*.

Last, but not least, I apologize to my many colleagues that have studied additional genes and epigenetic factors that can influence gap junction function, such as hemi-channels and annexins; other genes, such as pseudo-connexins, which can influence gap junction expression and the translation of connexin messages; and those genes that alter the functionality of connexin genes by phosphorylation/de-phosphorylation of the connexin proteins.

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
