# Peer review of "A Personal Scientific Journey—Looking Back at My Journey in Science: How DNA Damage and Repair Led to the Role of Gap Junctional Intercellular Communication, Mechanisms of Tumor Promotion, Human Adult Stem Cells, “Cancer Stem Cells”, Two Types of “Cancer Stem Cells” and the Modulation of Human Diseases by Epigenetic Toxins/Toxicants, Nutrition and Diets"

_cancers, 2025, doi:10.3390/cancers17162647_

Round 1
Reviewer 1 Report (Previous Reviewer 1)
Comments and Suggestions for Authors
This is a revised version of an interesting Viewpoint manuscript. Dr. Trosko was very responsive to the critiques and I have no additional concerns.
Reviewer 2 Report (Previous Reviewer 3)
Comments and Suggestions for Authors
In this revised manuscript, the author provides a nice summary of his research career and the unexpected events and discoveries that shaped his research career. This information will be useful to interested readers. I do not have any additional concerns with the manuscript. The revision has addressed all previous concerns.
This manuscript is a resubmission of an earlier submission. The following is a list of the peer review reports and author responses from that submission.
Round 1
Reviewer 1 Report
Comments and Suggestions for Authors
This viewpoint manuscript reflects the author's research career and how it is intertwined with understanding the molecular basis for carcinogenesis and roles for gap junctional intercellular communication in regulating orderly cell division and DNA repair. It presents a unique and personal take on the history of the field and will be of interest to others in the field.
In general, the manuscript is well written and is an interesting read. The author should consider dividing the manuscript into multiple sections to help readers follow to overall flow of the story.
Reviewer 2 Report
Comments and Suggestions for Authors
Dr. Trosko vividly illustrates his 50+ year scientific journey in this captivating article, highlighting numerous milestones across various fields, such as DNA damage, iPS cells, gap junctions, and cisplatin. I particularly enjoyed how Dr. Trosko uses road signs to metaphorically describe the scientific journey, making it relatable to our own life experiences. This beautifully written piece is a delightful read and would be a valuable addition to the Cancers journal, attracting significant interest and inspiring young scientists in their pursuits.
Reviewer 3 Report
Comments and Suggestions for Authors
In this manuscript, the author summarizes his career in science and highlights the important findings as well as unexpected detours. This article will be of interest to students and others interested in the field of DNA repair and carcinogenesis and the path of a scientist in this field.
There are some editing that could be done to improve the readability. Some of these are noted below:
1) Line 52: damage, not damaged
2) Line 56-57: wording is awkward.
3) Line 66: cancer, not cancr
4) Lines 95-98: Do you man inhibit DNA repair instead of DNA damage?
5) Lines 103-105: Delete the person's name, or re-word?
6) Line 110: damage genomic DNA
7) Line 145: supported, not support
8) Line 155: remove the word had?
Reviewer 4 Report
Comments and Suggestions for Authors
The manuscript presents a personal, reflective narrative that bridges a scientific journey spanning over five decades with the evolution of biomedical research, particularly in the context of gap junctions, Stem cells and cancer. It provides a unique longitudinal view of a career in science, which is valuable for junior researchers and students looking for mentorship through storytelling.